# Opportunities and challenges within green spaces during COVID-19: Perspectives of visitors and managers in Maine, USA

**Alyssa Soucy**[1]*, **Elizabeth Pellecer Rivera**[2], **Natalie Siwek**[3], **Lucy Martin**[2], **Sarah Jackson**[4], **Gabrielle Venne**[1], **Augusta Stockman**[5], **Sandra De Urioste-Stone**[1]

**1** School of Forest Resources, University of Maine, Orono, Maine, United States of America, **2** Ecology and Environmental Sciences, University of Maine, Orono, Maine, United States of America, **3** Thomas H. Gosnell School of Life Sciences, Rochester Institute of Technology, Rochester, New York, United States of America, **4** Park Management & Conservation, Kansas State University, Manhattan, Kansas, United States of America, **5** Department of Sociology, Vassar College, Poughkeepsie, New York, United States of America

* alyssa.r.soucy@maine.edu

## Abstract

COVID-19 impacted, and continues to impact, green spaces across the world, altering visitation patterns, and presenting novel management challenges. As COVID-19 has evolved, the long-term implications on communication, management, and conflict as diverse people interact in green spaces remains uncertain. Our work responds to calls to consider diverse perspectives of individuals whose lives intersect with green spaces. Using a qualitative case study methodology, we explored the meanings and experiences of green space managers and visitors in the State of Maine, USA, during the COVID-19 pandemic between May 2021 and July 2023. We triangulated across five research projects including: phenomenological interviews of conservation practitioners, an online questionnaire of staff from a state conservation agency, and three surveys of visitors to green spaces across Maine. Taken together, our results highlight how COVID-19 increased the number and diversity of visitors to green spaces as a result of the outdoor visitation opportunities provided perceived as "safe" during the pandemic. While managers described the benefits from increased visitation on funding and legislative support, they also identified challenges and negative impacts to local ecology as a result of new and greater use. Our results have implications for communication and management for policy makers and natural resource managers who seek to maintain support for conservation goals and address visitor safety and well-being. Managers maintained flexibility in their decision-making to remain nimble and responsive to emerging opportunities and challenges associated with the pandemic. In addition, our results highlight that the scale of change on human behavior from COVID-19 offers a glimpse into what may be possible if that same level of urgency was applied to issues like climate change.

**Data availability statement:** All relevant data are within the manuscript and its Supporting Information files.

**Funding:** This work was supported by the National Science Foundation under the Grants No. 1824961, 1828466, 1849802; the USDA National Institute of Food and Agriculture, McIntire Stennis project number #ME0-42017 through the Maine Agricultural & Forest Experiment Station; and by a University of Maine Research Reinvestment Fund Rural Health and Wellbeing Grand Challenge grant. The funders had no role in study design, data collection and analysis, decision to publish, or preparation of the manuscript.

**Competing interests:** The authors have declared that no competing interests exist.

## Introduction

Green spaces (e.g., urban parks and playgrounds, wildlife centers, state and national parks, publicly managed lands) contribute to both social and ecological outcomes as they provide opportunities for nature access and conservation [1,2]. During the COVID-19 pandemic, green spaces became even more of a necessity as people sought outdoor experiences that allowed social distancing. The pandemic altered visitation to green spaces [3,4], resulting in increased visitation to some locations [5–7], yet decreased visitation to others due to travel restrictions [8–10]. Groups also experienced differential access to green spaces depending on their locations and distance to more suburban/rural areas [11]. Therefore, COVID-19 revealed both opportunities and challenges for management [12] as decision-makers quickly adapted to emerging policies, and shifting visitation patterns, while balancing sociocultural and ecological goals [11,13,14]. At the same time, the COVID-19 pandemic shed light on inequities within green spaces as a result of geographic, financial, and physical barriers that limit green space access [2,15,16].

The long-term implications of COVID-19 on the ways people interact with green spaces are continuing to evolve [17,18]. For example, will visitation to parks and protected areas increase, remain stable, or decrease as the concerns with COVID-19 continue to decline? Will heightened visitors' human-nature connections remain? Will managers face ongoing challenges as a result of shifting visitation given the new users of green spaces that resulted from the pandemic? A better understanding of values, norms, and experiences of diverse individuals interacting in green spaces is necessary to manage socio-cultural, equity, and ecological goals [17]. This research aimed to explore the perspectives and experiences of natural resource managers and visitors as they interacted with green spaces in Maine during the COVID-19 pandemic. We brought together diverse perspectives to triangulate across experiences and understand areas of similarity and difference using a case study methodology. Specifically, we sought to understand the role, if any, that COVID-19 has played in how individuals experience green spaces, the perceived impacts of COVID-19 on these interactions, and coping mechanisms participants used to address those impacts. While previous work has investigated perceptions of visitors [19], and natural resource managers [20] during the COVID-19 pandemic, our case study draws on diverse perspectives over a period of two years. Our study responds to calls to address communication and management implications within green spaces as the pandemic has evolved [17,18,21–24]. We also further explore the potential long-term implications of COVID-19 on use and management of green spaces in Maine based on diverse perspectives, while drawing on lessons from the pandemic that may have implications on how we think and respond to other disturbances like climate change or an emerging epidemic.

## Literature review

### People and green spaces

People experience green spaces in both positive and negative ways [1,25,26]. From rural to urban contexts, green spaces can provide a positive intentional pause, or even welcomed interruption, for individuals to connect with the environment [27]. Green space exposure via walking or hiking, meditation and reflection, etc. can foster physical, psychological, cognitive, and emotional benefits [25,28–30]. In turn, connection with nature can also encourage stewardship behaviors that promote ecological benefits [31] However, experiences, historical relationships with land (i.e., exclusion from land), sociocultural practices, family roles, sense of belonging, experiences with discrimination, and resource availability informs how groups engage with the outdoors and the types of experiences they derive [32,33]. Individuals may

feel anxious or uncertain concerning the use of green spaces, which can ultimately inhibit the benefits of green spaces and make people feel unwelcomed [34,35]. In other words, the benefits of outdoor recreation have not been equitably shared across all communities [33]. Factors such as race and ethnicity, economic status, and gender have significantly impacted the extent to which structural, interpersonal, and intrapersonal barriers influence experiences in the outdoors [36–39]. For example, during COVID-19, the majority of individual those that participated in outdoor recreation in the U.S. – including those that were new to outdoor recreation – identified as being white, highlighting inequities in outdoor recreation and access [39].

Perceptions of risk and values can help to explain individual attitudes and behaviors as people differentially experience green spaces [40,41]. Risk perceptions [42], or personal judgments regarding the severity or likelihood of a threat [43], can determine how people evaluate and respond to hazards, such as COVID-19 [2,21], while values act as orienting beliefs which can steer behaviors [40,44]. A diversity of people engage with green spaces, whether through a management capacity (i.e., natural resource managers) or via engagement and outdoor recreation (i.e., nature-based tourists, residents, etc.) [45,46]. Therefore, their individual and collective values and perceptions need to be considered as they come together within shared spaces [45–47]. Natural resource managers must balance the varied values and perceptions [48–50] to implement managerial actions that thoughtfully consider issues of diversity, access, equity, and inclusion [29,51,52]. As COVID-19 has resulted in increasing visitation [5,53] and new groups of people engaging with the outdoors [20], these changes complicate existing values and risk perceptions [22].

Sense of place is an important concept within green spaces literature, helping to elucidate how individuals form relationships with their environment. Sense of place is a multifaceted concept composed of place attachment, and place meanings which formulate symbolic, meaningful experiences for individuals [54]. Specifically, place attachment is composed of place identity and place dependence. Place identity is an emotional attachment related to connectedness to place, often tied to personal or cultural meanings [55], while place dependence is a functional attachment related to the place providing necessary experiences [56]. Sense of place significantly influences not only individuals' experiences within green spaces, but also their trust in the management of these areas [57], motivations for recreation visits and satisfaction with their visits [58], and willingness to travel to or advocate for certain locations [59]. For example, studies have shown that strong place identity fosters trust in park authorities and stewardship, while high place dependence enhances satisfaction and loyalty to a specific green space [60,61].

## Emergence of COVID-19 in the context of green spaces

The COVID-19 pandemic, which emerged in Wuhan, China in late 2019, has so far caused over 6 million deaths worldwide, creating financial, mental, and physical health challenges across the globe [62]. Restrictions varied across regions, governments, and the urban to rural spectrum [4,63–65]. Global and national health recommendations, though varied in application, consistently purported social distancing, masking, reducing group sizes, and limiting travel as mitigation measures [64]. These restrictions caused unprecedented recreation facility closures and stay-at-home orders, which limited individuals' access to physical and stress-relieving activities, particularly those that were indoors [2,65].

While some green spaces experienced an increase in visitation during the first year of the pandemic [2,4,6,18,62], others experienced decreased visitation at the start of the pandemic due to mobility restrictions and park closures [66]. Closures in particular were significantly detrimental to parks globally as they are reliant on tourism for financial stability [4]. On a national scale, restrictions varied based on location, and visitors responded in kind — in

Maine for example, stricter pandemic protocols caused a 27% decline in visitation from 2019 to 2020 while states with fewer pandemic restrictions such as New Hampshire experienced an influx of often non-compliant visitors and over-congestion [24]. The high demand for outdoor recreation was due in part to the perceived safety that socially engaging in the outdoors presented in comparison to indoor gatherings, a sentiment that was encouraged by many state and local governments, and the lack of indoor gathering opportunities at the beginning of the pandemic [2,4,64,67]. In July of 2020, visitation to national parks was up 335% in just the three months since May, especially national parks near city centers [3]. In accordance with this heightened visitation and despite expanded testing and increased masking requirements, COVID-19 cases continued to increase into Fall 2020. Since 2020, case studies provide a snapshot in time, and often highlight the complex ways individuals differentially interacted with green spaces throughout the pandemic [18,24,68]. Overall, studies find that outdoor recreation participants changed their habits, recreating closer to home and engaging in activities that could be done while social distancing [2,22,64,69]. In some places, those who were already accustomed to using outdoor spaces sought out their use more frequently [67], while new users began to use the outdoors, yet lacked prior knowledge regarding norms and practices [21,22,67]. In national parks, reduced park staff and increased visitation during COVID-19 led to increases in poaching and illegal activities (i.e., littering, unregulated camping/hiking/fishing, and public urination/defecation due to reduced visitor monitoring capacity and access to public facilities) [3,4,64]. Increased visitation also raised concerned regarding trampling of sensitive flora or widening of trails as individuals adhere to social distancing measures – all of which can result in ecological damage [3].

Similarly, visitation to dispersed and remote recreation areas saw an increase earlier in the summer, while visitation to more developed outdoor recreation spaces peaked later in the summer, suggesting that visitors sought more remote locations early in the pandemic as they searched for safe ways to recreate [24]. Place-specific restrictions, and unique characteristics of green spaces (e.g., urban versus rural; type of ecosystems) [18,70], necessitates an in-depth understanding of impacts, experiences, and coping mechanisms grounded in a specific place. In doing so, results can address management concerns associated with COVID-19 and green spaces. For example, previous research identified the unique challenges and opportunities to natural resource managers associated with COVID-19, and developed several potential solutions including: implementing limits on visitation in certain areas to combat negative ecological impacts from increased visitation [24], and providing education to new or inexperienced participants who discovered green spaces during the pandemic to increase support for conservation [27].

This case study provides a novel and innovative contribution to research on COVID-19 and green spaces by examining diverse perceptions and pandemic-related impacts through the lens of place attachment within a rural region. Existing studies have highlighted changes in visitation during COVID-19 and its benefits for mental and physical health; however, this research distinguishes itself by focusing on Maine, a state characterized by its expansive green spaces and its reliance on tourism. Maine offers a compelling lens for understanding how place attachment—a sense of identity, connection, and emotional bond to specific locations—influenced responses to the pandemic and patterns of use and stewardship. By incorporating diverse perspectives from green space managers and visitors, this study captures the layered meanings and values associated with these spaces during the pandemic, offering transferable insights for other rural, tourism-dependent regions. Furthermore, the study's snapshot approach documents specific moments in time, complementing longitudinal research by revealing how place attachment evolved under crisis conditions [20]. These findings underscore the importance of integrating place-based values into adaptive management strategies to

sustain green spaces into the future. This research provides a critical framework for policy-makers and resource managers to balance ecological preservation with public access, while fostering meaningful connections that enhance both individual well-being and long-term conservation goals.

## Study site description

### Maine green spaces

Maine, located in the Northeast, USA, is approximately 91,646 km², of which 20% is currently conserved [71]. Conservation in Maine includes landscape-scale working forest easements, protected wetlands, productive farmlands and woodlots, working waterfronts, recreational uses, and areas encompassing Maine's natural heritage [72]. In addition, a diversity of land-owners with varied objectives work across the state (e.g., federal and state management, private conservation management, nonprofit organization management of easements and trusts, and Wabanaki tribal lands) (Fig 1). These groups balance ecological goals (e.g., habitat restoration, wildlife management, etc.), socio-cultural goals (e.g., green space access and out-door recreation, indigenous stewardship and subsistence priorities, etc.) and economic goals (e.g., timber harvesting and management). For example, while 94% of Maine's forested land is private, public recreation is allotted in more than half of this land. Additionally, the Bureau of Parks and Lands (BPL) manages Maine's Public Reserved Lands for primarily ecological ben-efits, and state parks primarily to provide green space access and outdoor recreation opportu-nities to visitors.

Maine is able to draw in visitors year-round due to the diversity of four-season outdoor recreation opportunities including hiking, swimming, camping, skiing, etc. Maine residents also have deep connections to the environment and natural resources, which are intimately tied to community identity, quality of place, and economic dependencies on natural resources [71,73]. Due to the state's diversity and availability of green spaces and activities, sense of place is strong among both local and out-of-state visitors to Maine's public lands [59].

### COVID-19 in Maine: Emergence, policies, and impacts to green spaces

Maine's first recorded case of COVID-19 occurred on March 12, 2020, nearly two months after the first US case on January 20. Three days after this first reported case, Governor Janet Mills declared a Civil State of Emergency. By the end of March 2020, all non-essential state businesses were closed, residents were directed to stay home, and nearly a dozen state parks were temporarily shut to visitors until April 8, 2020. Cases continued to rise in Maine, but as the summer tourism season approached, efforts were made to open up the state. In June 2020, out-of-state visitors who met either the fourteen-day quarantine or negative test requirement under the new Keep Maine Healthy plan could enter; by the beginning of July 2020, travelers from Connecticut, New York, New Jersey, and later Massachusetts were exempt from even these restrictions (Source: https://www.maine.gov/covid19/timeline).

Maine experienced first-hand the national trend of heightened interest in parks and other green spaces [2,4,67]. With fewer seasonal staff members available, parks had to prioritize the services that could be funded [3]. From May 1st to August 15th of 2020, as Maine opened back up, the Plan Your Visit section of park websites nationwide saw an 11% increase from the same period in 2019 [3]. Maine recorded its highest single day increase since the beginning of the pandemic on November 3, 2020, and by November 13, 2020, Governor Mills had back-tracked all of the state exemptions for visitors. Acadia National Park experienced a rebound in their visitation in 2021 and 2022, resulting in record-breaking visitation in 2021 (IRMA: https://irma.nps.gov/Stats/). Therefore, staff had to shift their focus to visitor management,

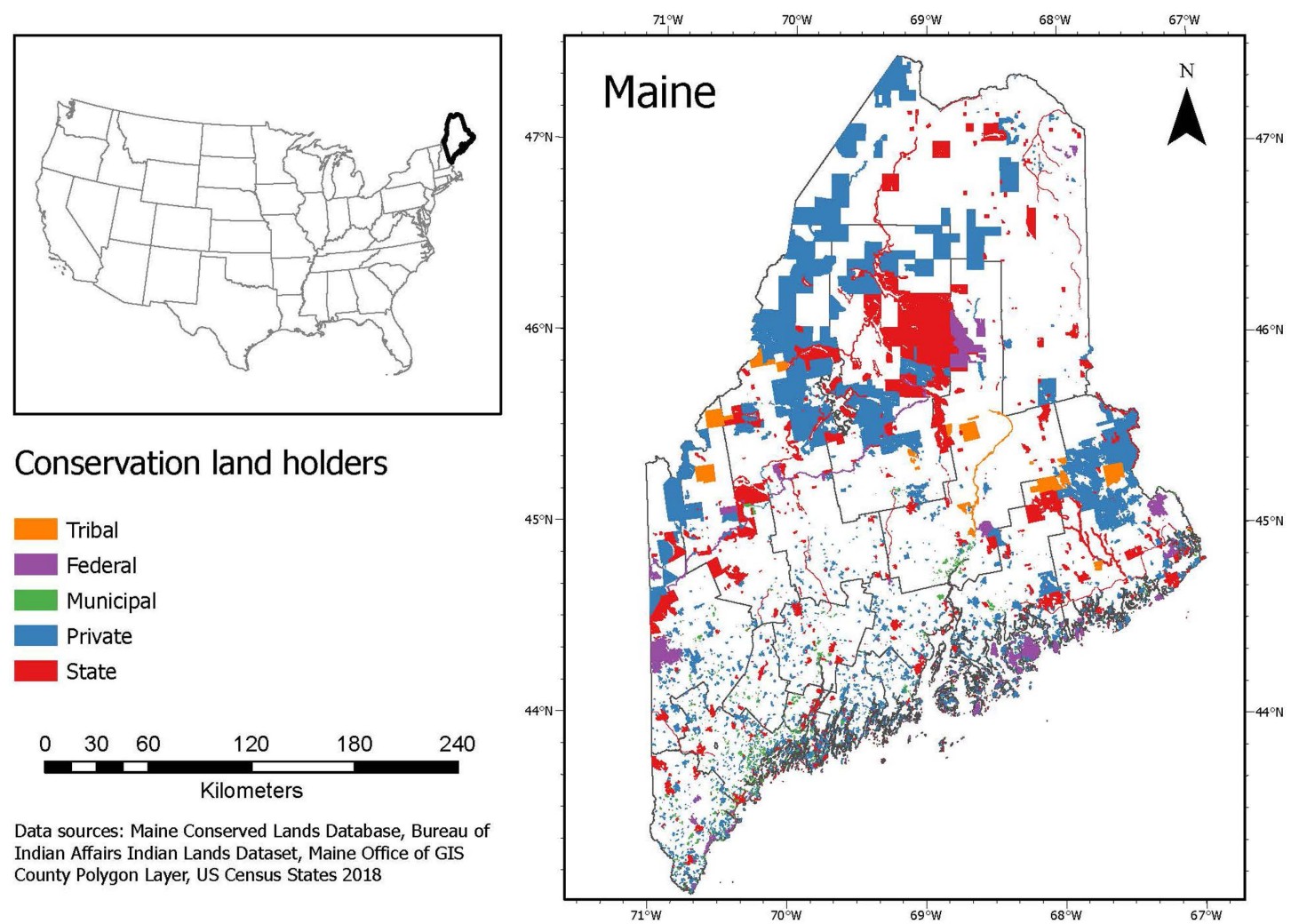

**Fig 1. Study site highlighting the different actors involved in green space management in Maine.**

leveraging social media to engage visitors with the park virtually and relay information on safe practices [3,27]. This increase in visitation and travel to rural areas is especially important given the close proximity of Maine to more urban areas such as New York City, Boston, and Philadelphia. The vaccine began to be available at the start of 2021, first to emergency personnel and residents aged 70 + in January, then to those 50 + in March, and finally to all residents 16 + in April. By May 7, over half of those eligible had been fully vaccinated. On June 30, 2021, Governor Mills finally lifted the State of Emergency (Source: https://www.maine.gov/covid19/timeline).

## Methodology

### Description of the overarching case study methodology

We used a single holistic case study methodology [74–76] that incorporates data from semi-structured interviews and questionnaires to gain a deep understanding of perspectives and experiences of Maine natural resource managers (i.e., conservation practitioners and staff from a conservation state agency) and visitors during the COVID-19 pandemic between

2021 and 2023 [77,78]. We simultaneously identified the common variables and constructs for comparison across quantitative studies, while exploring the relevant COVID-19 related themes in the qualitative studies. Through in-depth team discussions, data analysis, and data visualizations (e.g., word clouds, tables, and graphs) we triangulated across participants and methods [78,79] to understand areas of convergence and divergence. We focused on the key survey constructs which helped to illuminate the impact of COVID-19 on green space experiences and associated qualitative concepts that emerged to co-identify three key themes that all studies could engage with. By bringing together data collection methods over the course of several years in the evolution of the pandemic, we explored changes in assessing impacts, behaviors, and experiences.

## Data generation/collection methods and analysis processes

We collectively examined the results of five projects to understand various participant perspectives on the impacts of the pandemic on the use and management of green spaces in Maine (Table 1). Study names indicate participants (M for managers; V for visitors or outdoor recreationists), method (I for interview; Q for questionnaire), and the year data collection began. Study QM-21 is an online questionnaire [80] of staff from a Maine conservation state agency to explore the impacts of the COVID-19 pandemic on visitor management collected between April 27th to May 10th, 2021 [81]. Study IM-21 is a qualitative study to explore the experiences of conservation and stewardship practitioners, who are responsible for managing Maine's green spaces [82], based on in-depth semi-structured interviews collected between August 9th, 2021 and March 9th, 2022 [83,84]. Researchers conducted three visitor surveys between 2021 and 2023 to measure COVID-19 risk perceptions [85,86], place attachment [87], and travel behavior using 5-item Likert scales (e.g., 1 = 'strongly disagree' and 5 = 'strongly agree'). Specifically, Study QV-21 consists of an online survey [80] distributed to

**Table 1. Case study methods table–Research Projects that collected data on the COVID-19 pandemic as associated with green spaces in Maine.**

|  | 2021 |  | 2022 |  | 2023 |
|---|---|---|---|---|---|
| **Time period of data collection** | April 27th, 2021 - May 10th, 2021 | October 18th, 2021 – November 6th, 2021 | August 9th, 2021 - March 29th, 2022 | June 1st, 2022 - November 30th, 2022 | October 1st, 2022 - November 30th, 2022; March 1st, 2023 - May 1st, 2023 |
| **Study name** | QM-21 | QV-21 | IM-21 | QV-22 | QV-23 |
| **Participants (n)** | Conservation state agency staff (61) | Out-of-state visitors to Maine (407) | Conservation and stewardship practitioners (e.g., non-profit directors, tribal governments, state and federal government officials) in Maine (21) | Visitors to Acadia National Park (856) | Bureau of Parks and Lands recreationists in Maine (735) |
| **Research Goal** | Understand the impacts that the COVID-19 pandemic has had on visitations and natural resource management | Assess influences in the travel decision-making process of out-of-state visitors to Maine during 2020 and/or 2021 | Explore the impacts of COVID-19 on participants' experiences | Assess the impacts of the COVID-19 pandemic on risk perceptions and behaviors of visitors | Assess the impacts of COVID-19 pandemic on participation in outdoor recreation. |
| **Data Generation/ Data Collection** | Online survey | Online survey | Semi-structured interviews | Mixed mode survey | Online survey |
| **Data Analysis/ Software** | Thematic analysis of open-ended responses and cross-tabs/ SPSS 27.0 | Descriptive statistics/ SPSS 28.0 | Interpretative phenomenological analysis (IPA)/NVivo | Descriptive statistic analysis/ SPSS 28.0 | Descriptive statistic analysis/ SPSS 28.0 |
| **Trustworthiness/ Quality Assurance** | Instrument pretesting; use of previously tested scales | Instrument pretesting; use of previously tested scales | Peer debriefing Reflective journaling Prolonged engagement Member checking | Instrument pretesting; use of previously tested scales; non-response bias | Instrument pretesting; use of previously tested scales; non-response bias |

out-of-state visitors (CT, NC, MA, FL, NH, PA, VT, RI, NY) that had traveled to Maine in 2020 and/or 2021 collected between October 19th through November 6th, 2021 (S1 Table). Study QV-22 is a mixed mode survey [80] of Acadia National Park visitors collected between June 1st and November 30th, 2022 (S2 Table). Finally, Study QV-23 is a survey of Maine outdoor recreationists from two data collection periods including between October 1st, 2022 and November 30th, 2022 as well as March 1st to May 1st, 2023 (S3 Table). We ensured quality assurance in the qualitative study by keeping a reflective journal [88], remaining reflective of the researchers' roles [83], engaging in de-briefing [88], and seeking in-depth understanding through prolonged engagement [89]. We ensured quality assurance in the quantitative studies by using previously established scales [90], pre-testing the instruments [91], testing for validity of scales [92], and checking for non-response bias [93]. The University of Maine Institutional Review Board (IRB) for research on human subjects approved all studies. The IRB approved oral informed consent, given the structure of the interviews and surveys. We use pseudonyms to protect the privacy of all participants.

## Results

Across all studies participants were primarily white, had a range of educational experiences with most visitors receiving up to a 2-year degree and managers exceeding a 2-year degree, and had diverse ratios of male to female participants (Table 2). Participants noted and differentially experienced several key impacts as a result of COVID-19. These impacts included: shifts in visitation (e.g., visitors recreating at different times, increases in visitation, visitors newly discovering and reconnecting with the outdoors), ecological degradation due to increased visitation, and changes in the ways COVID-19 exacerbated and influenced other local to global issues. In turn, participants responded in diverse ways as they formed complex meanings around the new ways society interacted with green spaces. We conducted several cycles of analysis across all studies, comparing the similarities and differences in the data across perspectives and changes through time. We identified three themes, which focus on the categories that overlapped across all studies, and illustrate the diversity of experiences and responses: (1) Altered visitation patterns and public use implications, (2) Certain characteristics of Maine contribute to visitor perceptions of the state as a safe place to travel, and (3) Organizational challenges, opportunities, and response strategies. Throughout each theme we triangulate across studies to show connections across different participant groups.

### Theme 1: Altered visitation patterns and opportunities

Natural resource managers described increased visitation within Maine green spaces in response to COVID-19. A state agency director said, "COVID has increased the number of people that are using the outdoors…We've broken all-time records last year in both campgrounds and state parks day use" (Dana (All participant names have been changed to preserve anonymity), Study IM-21). At the beginning of the pandemic, conservation staff also noted the increase in participation in outdoor recreation, with 76% and 86% of participants finding a dramatic increase in the number of visitors across state parks and public reserved lands, respectively (Study QM-21). Visitor surveys also supported the propensity for outdoor recreation, as the majority (59%) of participants indicated that they primarily traveled to Maine to recreate (Study QV-21). Additionally, nearly 30% of outdoor recreationists reported an increase in outdoor recreation participation during the pandemic (Study QV-23).

With increased visitation, managers experienced both challenges and benefits as they sought to balance diverse ecological, social, and equity-oriented goals with green spaces.

**Table 2. Combined socio-demographics.**

| | Study QM-21 (n = 61) | Study QV-21 (n = 407) | Study IM-21 (n = 21) | Study QV-22 (n = 856) | Study QV-23 (n = 735) |
|---|---|---|---|---|---|
| **Ethnicity*** | | | | | |
| Black | – | 41 (11.8%) | 0 (0%) | 6 (0.7%) | 5 (0.8%) |
| Indigenous or Alaska Native | – | 8 (2.3%) | 3 (14.3%) | 8 (0.9%) | 9 (1.5%) |
| Hispanic/Latinx | – | 27 (7.8%) | 0 (0%) | 17 (1.9%) | 4 (0.6%) |
| Native Hawaiian or Pacific Islander | – | 2 (0.6%) | 0 (0%) | 1 (0.1%) | 1 (0.1%) |
| White | – | 255 (73.5%) | 18 (85.7%) | 769 (89.8%) | 527 (88.72%) |
| Asian | – | 14 (4.0%) | 0 (0%) | 41 (4.7%) | 3 (0.5%) |
| Middle Eastern | – | 0 (0%) | 0 (0%) | 0 (0%) | 2 (0.3%) |
| **Education** | | | | | |
| Less than High School | 0 (0%) | 12 (3.7%) | 0 (0%) | 1 (0.1%) | 3 (0.5%) |
| High School grad | 1 (2%) | 93 (28.8%) | 0 (0%) | 31 (3.6%) | 39 (6.8%) |
| Some college | 4 (7%) | 72 (22.3%) | 0 (0%) | 67 (7.9%) | 90 (15.7%) |
| 2-year degree | 6 (11%) | 51 (15.8%) | 0 (0%) | 47 (5.5%) | 47 (8.2%) |
| 4-year degree | 26 (48%) | 70 (21.7%) | 9 (42.9%) | 309 (36.4%) | 191 (33.3%) |
| Professional degree | 4 (7%) | 0 (0%) | 1 (4.7%) | 0 (0%) | 22 (3.8%) |
| Master's degree | 10 (19%) | 20 (6.2%) | 8 (38.1%) | 281 (33.1%) | 147 (25.6%) |
| Doctorate | 1 (2%) | 5 (1.5%) | 3 (14.3%) | 105 (12.4%) | 25 (4.4%) |
| Technical College | 0 (0%) | 0 (0%) | 0 (0%) | 0 (0%) | 10 (1.4%) |
| **Gender** | | | | | |
| Male | 40 (74%) | 125 (38.7%) | 10 (47.6%) | 375 (43.8%) | 364 (65.0%) |
| Female | 9 (17%) | 197 (61.0%) | 11 (52.4%) | 462 (54.0%) | 193 (34.5%) |
| Non-binary/other | 1 (2%) | 0 (0%) | 0 (0%) | 0 (0%) | 3 (0.5%) |

*Values do not add up to 100 given this is a multiple selection question.

*Note: we used listwise deletion; therefore, not all numbers add up to total N values and percentages relate to the valid percent*

Eighty percent of conservation staff surveyed expressed concern for limited personnel, visitor health, and staff safety across state parks and public lands in the summer of 2021 (Study QM-21). Specifically, conservation staff noticed the increase in new visitors on public lands,

> An increase in visitors who are not knowledgeable in the conditions…while visiting Maine Public Lands. Big increase in out-of-state visitors to more remote parts of the state. These visitors often have higher expectations of the accommodations/ situations they will be in while visiting these areas.

This manager describes the associated challenges of new visitors experiencing green spaces for the first time as they seek to provide high quality user experiences (Study QM-21). Visitor surveys also supported these manager perceptions, as 37% and 72% of out-of-state visitors between 2020 and 2021 were first time visitors to Maine and Acadia, respectively (Study QV-21). Similarly, 62% of those who visited Acadia National Park in the tourist season of 2022 were first time visitors (Study QV-22).

While people were drawn into Maine's scenic beauty, managers grew concerned over the increased visitation numbers being at odds with the need to minimize ecological damage (i.e., vandalizing, littering). A state agency director reflected on the overcrowding at state parks that occurred in March and April of 2020:

> One day we had 2,000 people showing up and 1 person on staff…and we really [didn't] know enough to be making these difficult decisions about whether [to shut down], so we ended up closing state parks for about a month…We've seen some places where there's too many people for the resource and we haven't really figured out the most effective way to deal with that yet…we were just getting a lot of people camping up there, and partying up there, and leaving trash, and burning trees. (Dana, Study IM-21)

For managers, increased use brought up questions regarding balancing equitable and inclusive visitation opportunities with ecological goals, such as whether or not to institute a campground reservation system at some heavily trafficked areas. As a land trust manager says, "There's also been a lot more use on our preserves so we're also seeing the downside…and the issues that go along with having a lot more people using our preserves." (Emerson, Study IM-22). A the same time, managers also recognized that this was an opportunity to assess access to these green spaces and the values they considered important to conserve, as a non-profit manager says, "and we need to do a better job at not placing places in glass boxes" (Jordan, Study IM-22), as they discussed ensuring diverse people remain connected and feel that they belong within green spaces across the region. A possible solution emerged as Reese described the ways they balance ecological goals with public access:

> 20% of our landbase is ecological reserve…[with] few trails that pass through… And, those areas were specifically chosen based on having special management areas, unique ecosystems, and in some cases resources that couldn't withstand a lot of high intensity recreational use. (Reese, Study IM-22)

In designating areas for ecological management and others for access and recreation, a balanced approach is one way managers described dealing with the negative impacts associated with increased visitation.

Visitors similarly expressed concerns regarding crowding, and identified perceived proximity to other visitors as a primary cause of changing their behaviors (e.g., visiting earlier or later in the day, going to a different area of the park, changing preferred activity, ending their trip early, etc.) (Study QV-22). In addition, visitors noted that accessibility to campsites and other activities was much more limited, as participants commented, "Biggest change since 2020 is availability of camping sites at the state parks. Everything has to be booked well in advance and everything has been discovered," (Study QV-23) and,

> My experience has been devastating because everyone is going to state parks now and have no respect with noise…We can't get our campsites anymore even if we change the dates… we are getting squished out of our favorite time. (Study QV-23)

Visitors expressed sadness at the inability to compete with increased visitation and visit beloved green spaces in Maine. Despite the challenges of crowding, safety, and personnel management, several benefits emerged. Natural resource managers reflected on opportunities for bringing together people in green spaces. A state employee said,

> The pandemic has shifted everything so much and so many people are going outside and engaging…I don't think we would have gotten to where we are right now without the pandemic. And so I think the pandemic has catapulted us forward with outdoor engagement. (Drew, Study IM-21)

Drew expresses a hope to continue to connect people with green spaces and grow outdoor engagement. As a result of increased engagement, support in the form of funding and legislation surfaced during the pandemic. A state agency director described the positives of increased engagement, "We've been able to effectively communicate with the legislators about the importance of the outdoors…made our work more successful because of the funding opportunities." (Dana, Study IM-2). Meanwhile, a land trust manager said, "We had a huge surge in popularity at our preserves and as a result of that a lot more supporters, which has been really great." (Emerson, Study IM-21). Therefore, despite the burdens of crowding, and ecological damage as a result, increased use during the pandemic shined a light on the benefits of green spaces among visitors, funding agencies, and policy-makers.

## Theme 2: Certain characteristics of Maine contributed to visitor perceptions of the state as a safe place to travel

Visitors during the pandemic highly valued precautionary measures (e.g., social distancing, mask wearing, high vaccination rates) and a low risk of contracting COVID-19 as influential in both selecting their travel plans and the overall quality of their travel experience (Table 3). Results from the visitor surveys suggest the increase in visitation to Maine green spaces during COVID-19 may therefore have been a result of perceived trust and safety, as well as a high sense of place attachment to Maine that became increasingly important during the pandemic (Table 3). Early in the pandemic, in particular, visitors noted that they were worried about COVID-19 (Study QV-21) and influenced to travel to Maine due to perceptions of safety around COVID-19 (Study QV-21 and Study QV-22), yet, concern and worry decreased across the years between visitors' studies (Table 3). Of course, participants still noted the role of the pandemic in influencing their decisions; as a visitor wrote, "By choosing outdoor activities and picking trails that were not as crowded, we felt that we were able to enjoy nature without having to worry about social distancing" (Study QV-22). Therefore, participants held diverse perceptions which factored into their decision-making depending on the time and location of their experiences.

**Table 3. Mean values for selected items for visitor study questionnaires.**

| Item and measures | Study | | |
|---|---|---|---|
| | QV-21 (N = 407) | QV-22 (N = 856) | QV-23 (N = 735) |
| **Place Identity**[*] | 3.52 (0.75) | 3.49 (0.72) | N/A |
| **Place Dependence**[**] | 3.37 (0.86) | 2.95 (0.77) | N/A |
| **Travel and COVID-19** | **3.35 (0.64)** | **3.35 (0.5445)** | **3.09 (0.6158)** |
| Traveling to rural destinations makes me feel safe during the pandemic | 3.66 (1.00) | 3.64 (0.963) | 3.78 (1.011) |
| I am not concerned with safety when choosing to travel to destinations | 2.65 (1.27) | 2.49 (1.155) | 3.21 (1.256) |
| Traveling to areas that have higher rates of COVID-19 vaccinations makes me feel safer | 3.21 (1.29) | 3.47 (1.240) | 3.02 (1.313) |
| Traveling to nature-based destinations (such as a park) makes me feel safe during the pandemic | 3.81 (0.97) | 4.04 (0.900) | 3.86 (1.042) |
| I am personally worried about COVID-19 | 3.52/1.228 | 3.28 (1.188) | 2.87 (1.305) |

[*]Items consisted of 'I feel Maine is part of me'; 'I feel very attached to Maine'; 'I identify strongly with Maine'; 'Maine means a lot to me'; 'Maine is very special to me'; 'Visiting Maine says a lot about who I am'.

[**]Items consisted of 'I get more satisfaction out of visiting…'; 'I would not substitute..'; 'Maine is the best place…'; 'No other place can compare to Maine'

Despite these changing and complex perceptions around safety, across visitor studies in both 2021 and 2022, participants indicated a high sense of place identity and dependence as they felt connected and attached to Maine (Table 3). In particular, out-of-state visitors in the early pandemic expressed a high sense of place dependence (Study QV-21; Table 3), which also may have contributed to heightened visitation as visitors felt that Maine was the best place for what they wanted to experience. Participants expressed a perception of safety specifically around rural and nature-based destinations, with 67% of out-of-state visitors (Study QV-21) and 79% (Study QV-22) of visitors to Acadia National Park indicating that they felt safe traveling to nature-based destinations during the pandemic. Participants described their predilection towards Maine, as one visitor wrote, "I LOVVVVE Acadia. We chose Acadia last year because of the pandemic and felt safe being outdoors. Loved it so much we came back this year" (Study QV-22)! An out of state visitor wrote,

> We planned a picnic in our beloved Maine. We will be back for overnights once we can be assured this pandemic is in our rear view mirror. We trust Maine, we just don't trust others to do what is right. (Study QV-21)

These findings illustrate the value that visitors placed on not only safety, but also on their attachment and connections to Maine.

Managers noted the connections between perceptions of safety, sense of place, and increased visitation within green spaces across Maine. A land trust director said,

> We've also received a lot of new members who are finding our preserves for the first time because COVID made it so that people just like flocked to natural areas and parks, because it was one of the only like safe activities. (Emerson, Study IM-22)

Much like the visitors who choose to visit Maine for perceived safety, but continued to come back the next year, practitioners noticed that people discovered and reconnected with green spaces during the pandemic. A state agency director said, "when the indoors was closed people went outdoors. So I think people are in droves across the country are kind of connecting and reconnecting with the outdoors." (Dana, Study IM-22). The discovery of green spaces in Maine, therefore, has much to do with the rural nature-based opportunities which offer safety from crowds, as well as visitors' high sense of place. Managers see this as an opportunity to continue to leverage those place and nature connections, as a state agency director said,

> If we could keep all the people who are outside participating in outdoor activities right now, keep them connected, keep them engaged …so that when everything goes back to whatever the next normal is, they're still doing those things, they're still going hiking on the weekends, they're still fishing with their kids… they're still doing those things in 5, 10 years, it becomes part of their lifetime choice. (Drew, Study IM-22)

If those connections persist, managers remain hopeful that financial and social support for green space conservation will continue to grow.

### Theme 3: Organizational challenges, opportunities, and response strategies

In addition to altering visitation patterns, the beginning of the pandemic was characterized by uncertainty, given that limited knowledge about the disease was available. COVID-19 caused stress and anxiety in the staff and managers of Maine green spaces. A manager describes the

beginning of COVID-19 as the "most stressful couple months of my professional career," (Dana) as they faced difficult decisions around closing parks due to staff shortages and uncertainty around disease transmission. In 2021, just one year after the pandemic started, Maine state conservation agency staff considered COVID-19 (56%) among the top three factors that most affected their work over the past 5 years, just behind staffing (74%) and budget (72%) (Study QM-21). Managers acknowledged how COVID-19 was very much tied to these staffing and financial challenges. One manager wrote,

> Covid 19 has affected the state parks financially and has increased our visitor use. Funding has always been lacking for State Parks and the parks continue to have a backlog of work that need to be done…The pandemic has driven our public use way up…The issues are underpaid staff, lack of staff positions. (Study QM-21)

Therefore, these pandemic challenges were only intensified by other local to global-scale issues already occurring. Managers recognized that COVID-19 "makes everything take longer" (Emerson, Study IM-22) and therefore impacted much of their collaborative approach to decision-making. Two managers additionally noted the supply chain disruptions in regards to forest management and stewardship (Study IM-22). These responses highlight how COVID-19 intersected with local to global economies, disrupting visitor management and stewardship in complex ways.

Managers also faced organizational issues around programming capabilities; however, despite the challenges of virtual programming, managers acknowledged the positive outcomes, as the virtuality and the restrictions placed on indoor spaces allowed them to diversify their programs and reach a wider audience. From the perspective of visitors, COVID-19 impacted information channels used and trusted. In particular, visitors indicated that they highly trusted the government (43.2%, Study QV-21; 59.3%, Study QV-22) and family/friends (53.5%, Study QV-21; 39.8%, Study QV-22) to receive information about COVID-19. A manager for a private landowner describes the challenges around communicating effectively with legislators and the public, "'Cause everything was Zoom…It's hard…to convey information in a good way between people that don't agree at all over a computer" (Quinn, Study IM-22). At the same time, managers faced challenges with programming, the director of a land trust describes the shifts,

> One of the biggest challenges has been our educational programming… At times [public programming and local school programs have been] completely impossible over the past year, and at other times our education person has had to do Zoom based nature lessons. (Reese, Study IM-22)

Therefore, managers' ability to directly communicate with visitors and policy-makers using traditional in-person information channels had to adapt to remote programming. The inability to communicate using in-person modalities created challenges for managers to engage with the public about important conservation issues, including visitation, as they had to reconsider their traditional strategies.

Further, the majority of Maine conservation agency staff (58%) expressed impacts in their efforts to manage climate change due to the pandemic. One Maine conservation agency staff member wrote, "The pandemic has taken away already limited time by creating new priorities for management" (Study QM-21). The longer tourism seasons created by climate change exacerbated staffing shortages that became even more difficult during the increased visitation as a result of COVID-19. As one Maine conservation agency staff member wrote,

> [the extended tourism season] creates a challenge when our seasonal staff is gone and our limited year round staff is stretched to cover the "off-season"…As the climate continues to warm staffing levels will need to change or the quality of our conserved areas will suffer and so will the visitor experience. (Study QM-21)

However, despite the challenges of this interconnectedness between COVID-19 and climate change, managers also expressed hope at the connections between these two global issues,

> We learned that people can change their behavior. People wear masks now all the time regularly, and we never would have even considered that. People have adapted to all kinds of things that none of us expected… I think that's super optimistic that people can make those changes. (Drew, IM-22)

As Drew describes, the scale of change on human behavior from COVID-19 offers a glimpse into what may be possible if that same level of urgency was applied to issues like climate change. Despite the challenges posed by the pandemic, managers maintained flexibility and adaptability in order to quickly respond, overcome, and leverage evolving changes and obstacles that emerged since 2020.

## Discussion

Our findings describe the impacts of COVID-19 on visitors and managers' experiences with green spaces. The diverse datasets allow an opportunity to reflect on areas of similarity and differences in opinions and experiences among participant groups and the ways individuals coped with the impacts of COVID-19. Participants identified the COVID-19 pandemic as a catalyst for increased visitation in green spaces in Maine. Rapidly changing guidelines, recommendations, and policies from national, state, and local governments influenced accessibility, while visitation levels altered accordingly. At the same time, managers described closing parks early in the pandemic, and then experiencing dramatic increases in later stages of the pandemic in 2020 and extending through 2022. Researchers reported similar visitation trends in other geographic locations in both urban [18,94] and rural settings [6,20].

Specific to the Northeast, US, Ferguson et al. (2022) found that COVID-19 increased outdoor visitation during the summer of 2020, with both in-state and out-of-state visitors exploring dispersed green space sites. Given the particular characteristics of COVID-19 disease transmission [95], Maine's rural characteristics, and the measures put in place to prevent infection (i.e., social distancing, mobility restrictions) [24,96], visitors perceived Maine green spaces as a safe place to travel. At the same time, visitors' high sense of place attachment towards Maine provided a unique opportunity for the state to experience increased and sustained visitation throughout the pandemic, particularly after the early restrictions and park closures. Research prior to COVID-19 suggests that visitors' place attachment in Maine was relatively lower in 2016 compared to the pandemic and post-pandemic [De Urioste-Stone, personal communication]. The COVID-19 pandemic therefore may have brought visitors with higher place attachments and/or bolstered place attachment for return visitors. As a result, supporting Maine's ability to draw in visitors. Despite participants' concern regarding the COVID-19 pandemic, for many, the benefits of visiting green spaces outweighed the risks. For example, Maine visitors appreciated the variety of nature-based opportunities that Maine offered, and the ability to not only recreate safely, but to do it in a place that they loved.

## Flexibility and leveraging opportunity as critical mechanisms to respond to COVID-19

As visitors altered their travel plans to reduce their COVID-19 risks, managers had to quickly determine policies to create equitable green space access while complying with state guidelines and upholding their management aims. Visitors made personal judgments to enter and leave spaces at certain times to reduce crowding and increase comfort levels. This is similar to previous research conducted across five different national parks [19]. The pandemic influenced both popular tourist destinations, and local parks and trails as people engaged in recreational activities both regionally across Maine as well as more locally. Similar to visitors, managers' experiences during the early pandemic were characterized by a focus on safety. Our results highlight feelings of anxiety, concern, and overwhelm among managers as they dealt with staffing shortages and policy changes, similar to previous research [3,19]. Green space managers adapted by maintaining flexibility, directing users to different areas within parks, closing areas of parks due to low staffing, and installing reservation systems such as the case in Acadia National Park. Green space managers elsewhere similarly remained nimble to changing norms and values regarding perceptions of benefits, safety, and crowding [3,19,53,97]. Managers of green spaces in Maine noted the complex ways in which COVID-19 both generated support and remained in conflict with ecological goals. Within Maine, natural resource managers described sustained heightened participation and engagement of green spaces a year and a half (Summer of 2021) after the start of COVID-19. Few studies have examined the longevity of visitation increases; however, Venter et al. (2021) similarly found increases in recreational activity despite an easing of lockdown measures during the first year of COVID-19 in Norway. Our results suggest that visitors' high sense of place may be contributing to sustained high visitation, which was catapulted by safety concerns around COVID-19. As indicated by both manager perspectives and visitor surveys, as visitors connected and reconnected with the outdoors, they discovered Maine and formed a strong sense of place attachment. In turn, our results highlight the power of leveraging the opportunities presented by COVID-19 challenges to garner support for green space conservation.

However, outdoor spaces saw many first-time and returning visitors due to the outdoors being a perceived safe space from COVID-19, a finding that is similar to previous research [62]. As managers described new users connecting with the outdoors – a finding that is similar to other studies in Northeast, US [98], and globally [20,24,27,99] – managers noted special considerations around communicating safety and expectations to groups that are unfamiliar with remote and rural green spaces. Previous research indicates that novice visitors may further complicate issues of communication and management, as these new visitor groups may have different motivations and expectations [14]. Our results suggest that visitors were more likely to travel alone during the pandemic as compared to previous surveys of Acadia National Park visitors prior to COVID-19 (e.g., [59,100–102]. The activities visitors engaged in (e.g., hiking, sightseeing, viewing wildlife) did not drastically differ from those visitors engaged in at Acadia National Park in previous surveys prior to COVID-19 (e.g., [59,100–102]; however, in 2021 and 2022 visitors reported nature photography as one of their top activities, replacing activities related to shopping and dining in surveys from previous years. This is unsurprising given restrictions and perceptions of safety related to indoor activities. Also, our results suggest that those surveyed in Acadia National Park during COVID-19 had similar socio-demographics (e.g., age distributions, household income, and education) compared to surveys conducted prior to COVID-19 (e.g., [59,100–102]; This suggests that while the percentage of first time visitors increased, their characteristics do not drastically differ from pre-COVID visitors. Our results therefore demonstrate that unlike previous research, and the perceptions

of managers around communicating with different first-time visitors, survey results do not indicate a drastically different visitor socio-demographic before and during the pandemic. Similarly, while the majority of literature often cites the challenges associated with understanding novice visitors that discovered green spaces during COVID-19 [20], our results also point to the benefits of these new supporters and their human-nature connections [17].

## Balancing diverse green space goals during COVID-19

COVID-19 made salient the balance between managing socio-cultural, equity, and ecological goals as visitation increased. Managers faced challenges associated with creating accessible and quality visitor experiences, alongside conserving ecosystems and minimizing ecological damage within shared spaces. These conflicts have been present in the green space literature for decades, and were exacerbated by COVID-19 [3,9,13,14,62]; however, our study showcases potential coping mechanisms from managers when faced with an unprecedented shock, like the pandemic. While managing increased use and the associated impacts (e.g., increased litter), managers also harnessed the increased participation and support generated from the COVID-19 pandemic to promote conservation goals. In doing so, they articulated a vision for green space conservation where benefits for people and nature are not mutually exclusive. Previous work has primarily focused on the negative impacts of increased visitor use on ecological goals during COVID-19 [20]; therefore, our results shed light on an optimistic co-existence whereby increased visitor use can also support ecological goals as some areas are managed for conservation while others are managed for visitor use and access.

Visitors' high sense of place identity with Maine highlights a potential opportunity for managers to communicate the importance of adhering to safety and use guidelines to conserve Maine's many socio-cultural and environmental values. Participants indicated a preference to spend time in areas that they loved and perceived as safe. Maine's rural environment makes it unique in respect to other COVID-19 and green space studies which have often focused on urban areas [6,103]. In studies that have focused on rural green spaces, similar findings suggest the important role of sense of place in contributing to increased funding and re-visitation [99]. Our results also indicate shifting perceptions among visitors, as the importance visitors placed on COVID-19 safety decreased over the years, despite a relatively stable sense of place. While COVID-19 concern may have dissipated among visitors, a high sense of place may continue to sustain heightened visitation to Maine as people discovered and rediscovered the state during the pandemic.

At the same time, COVID-19 illuminated Diversity, Equity, and Inclusion (DEI) concerns related to green space access and management in Maine. Managers acknowledged how achieving their dual goals of providing recreational opportunities and maintaining local ecology was challenged during the pandemic, forcing them to consider coping mechanisms that also had implications for DEI (e.g., considering online reservation systems that require internet access). Therefore, while green spaces provided an escape from COVID-19 for many, prior research has indicated that not all individuals had access to these spaces [15,25]. For some managers, COVID-19 created a space to consider the values their land actually conserved and the people who were able to connect with those spaces.

## Management implications and future planning

Our results have several implications for management in regards to communication, collaboration, and adaptation. Managers and visitors within green spaces will likely face similar challenges in the future when it comes to using shared spaces, and continuing to grow and

leverage visitation while meeting ecological goals [14], and ensuring equitable access [104]. Epidemics due to emerging infectious diseases will continue to occur [105,106], especially in regions that rely heavily on tourism to support the economy [21,107]. The impacts of emerging diseases (i.e., Ebola, H1N1, MERS) on tourism have drawn global attention over the last several years [108,109]. The ripple effects on economies, health, and ecosystems from emerging diseases can trigger unknown consequences. For example, increased visitation to green spaces from COVID-19 can also lead to health-related concerns, including an increase in mosquito-borne diseases [110], and tick-borne diseases [111] due to changing mobility patterns and novice users who may be unaware of these health risks.

Although COVID-19 presented a threat to global and local visitation at the global scale [96]; our study shows that at a local scale and in a medium term, the pandemic also presented diverse positive outcomes for visitors and conservation efforts. Our results point to the importance of leveraging existing human-nature connections in communication efforts to capitalize on the opportunities that emerged from COVID-19. Social and ecological goals can coexist when managers harness the power of human-nature connections in order to increase support and provide quality visitor experiences. Tools that were developed during the pandemic such as virtual programming are likely to be fundamental in this process of strengthening human-nature relations. To leverage the opportunity from increased support, managers must consider the possibility of future socio-ecological disturbances, like pandemics in their green space planning and monitoring [22,112]; for example, by considering a social carrying capacity threshold for areas to continue to balance diverse goals, as well as designing spaces that allow physical distancing – recommendations also identified in previous literature. At the same time, managers must continue to think at scales that transcend jurisdictional boundaries [19] - much like the coordination of regional conservation goals described by managers, they must also work together to develop visitor use strategies that unify messaging, leverage human-nature connections, disperse visitors when needed, and ensure local values are upheld when promoting outdoor recreation [10]. Given the hybrid nature of communication channels due to COVID-19, communications that leverage existing social networks that visitors likely trust, may be one such approach to get out messages around recreating safely and minimizing negative ecological impacts (e.g., maintaining six feet of distance, Leave No Trace principles, etc.) [113].

Managers' flexibility and adaptability can be translated to manage future emerging and unpredicted disturbances beyond epidemics, such as issues related to climate change [27]. As COVID-19 compounds climate change risk [114,115], the literature has increasingly focused on the intersection of COVID-19, climate change, and social vulnerability and adaptation [114,116]. Our results highlighted the interconnectedness of COVID-19 with other challenges, as participants mentioned (1) the possibility of addressing climate change by considering the lessons learned from COVID-19, and (2) the ways in which COVID-19 impacted their ability to respond to climate change. As managers described their optimism for climate change response given the rapid behavioral change during COVID-19, our results showcase how the pandemic offers an opportunity to learn about capacity building, resilience, and rapid behavioral response. Similar themes have been noted in the literature around sustainable development [117,118]; however, our results shed light on the perceptions of optimism among green space managers. What we learned from COVID-19 is that major changes can be accomplished quickly [117], and that COVID-19 will not be the only time we face compound risks [114]. Future research should continue to address the intersections of socio-ecological disturbances like climate change, COVID-19, and other health-related concerns as green space managers face increasingly complex challenges that require difficult adaptive decision-making.

## Limitations and future research

Our findings represent an in-depth case study of managers and visitors in Maine green spaces as they experienced and adapted to COVID-19. In certain contexts, our results may be transferable and applicable to understand diverse perspectives around COVID-19 and green spaces, as many individuals recreate together within shared spaces. While we included a diverse range of perspectives (e.g., managers from state, Tribal Nations, and federal groups; in-state and out-of-state visitors), we had to balance the breadth of perspectives with a depth of understanding. Therefore, future work would benefit from an inclusion of local residents – a group we did not account for in this case study and who would be directly impacted by green space management decisions, and visitation impacts. Local residents also have the ability to influence green space management via public input (i.e., supporting or opposing visitation restrictions), and therefore represent a missing actor in the study. Given the diversity of datasets (e.g., different scales/areas of focus, and different survey items), we were also limited in the extent to which we could easily make comparisons across our data and with past surveys conducted in the region. For example, Study QV-21 was not specific to Acadia National Park, but rather conducted as part of a larger regional study in Maine. Therefore, we made general comparisons where possible, but did not rely on tests of significant differences given the different questionnaire formats and slight differences in survey question formats. In addition, further work could explore more in-depth the values and norms of Maine visitors, building on the quantitative analysis in this study. Future research could build on our approach to triangulating different data sources to continue to identify similarities and differences in perceptions and experiences by members of diverse groups around the management of varied values within green spaces, and the ways in which disturbances such as the COVID-19 have impacted the balance of those values.

## Conclusion

The impacts resulting from the COVID-19 pandemic continue to be complex and evolving, spanning well beyond the end of the global health emergency. Our case study captured the perspectives from various visitor and manager groups across a two-year timespan. Pulling data and experiences from multiple studies, we developed an in-depth picture of how the impacts on green spaces evolved as the pandemic persisted, and the ways in which managers and visitors coped with emerging challenges and opportunities. Despite the challenges posed by the pandemic, managers' responses remained nimble and adaptive to changing impacts and obstacles that emerged since 2020. While COVID-19 exacerbated already existing challenges around balancing diverse management goals; ultimately, managers described the overwhelming support the pandemic catalyzed for green spaces across Maine. Within the state, the multitude of green spaces available for visitors alongside the diversity of activities allowed Maine to be perceived as a safe and entertaining destination to travel to, especially for visitors in Maine and the surrounding metropolitan areas. Maine provided a unique case study due to its popularity as a vacation destination for these metropolitan areas and possession of the only national park on the U.S. East Coast. In addition, Maine is an interesting case study given that visitors had a relatively high place attachment to the state as compared to pre-pandemic visitors. Therefore, the state's green spaces continued to see visitation, causing opportunities and challenges for managers already burdened by staff and resource restrictions. Over the course of the pandemic, a heightened sense of place persisted and Maine experienced sustained high visitation of green spaces. When we consider the future of green spaces in light of COVID-19, implications from this study may be transferable to other socio-ecological disturbances like climate change or future epidemics, and the mechanisms that can be used to cope with the

negative implications of these challenges. For many green spaces, a continued and renewed effort will need to be made to balance socio-cultural, equity, and ecological goals, especially as complex socio-ecological challenges continue to emerge.

## Supporting information

**S1 Table. Study QV-21 data table.** Raw data for the relevant data sets from QV-21. (DOCX)

**S2 Table. Study QV-22 data table.** Raw data for the relevant data sets from QV-22. (DOCX)

**S3 Table. Study QV-23 data table.** Raw data for the relevant data sets from QV-23. (DOCX)

## Acknowledgments

We would like to thank the participants who shared their experiences and opinions across all studies, including visitors and managers. We thank our colleagues, Valeria Briones and MacKenzie Conant for their contribution to data collection and idea formation.

## Author contributions

**Conceptualization:** Alyssa Soucy, Elizabeth Pellecer Rivera, Natalie Siwek, Lucy Martin, Sarah Jackson, Augusta Stockman, Sandra De Urioste-Stone.

**Data curation:** Alyssa Soucy, Elizabeth Pellecer Rivera, Lucy Martin, Sarah Jackson.

**Formal analysis:** Alyssa Soucy, Lucy Martin, Sarah Jackson.

**Funding acquisition:** Sandra De Urioste-Stone.

**Investigation:** Alyssa Soucy, Elizabeth Pellecer Rivera, Natalie Siwek, Lucy Martin, Sarah Jackson, Gabrielle Venne, Sandra De Urioste-Stone.

**Methodology:** Alyssa Soucy, Elizabeth Pellecer Rivera, Lucy Martin, Sarah Jackson, Augusta Stockman, Sandra De Urioste-Stone.

**Project administration:** Alyssa Soucy.

**Supervision:** Alyssa Soucy, Sandra De Urioste-Stone.

**Validation:** Alyssa Soucy.

**Writing – original draft:** Alyssa Soucy, Elizabeth Pellecer Rivera, Natalie Siwek, Lucy Martin, Sarah Jackson, Gabrielle Venne, Augusta Stockman.

**Writing – review & editing:** Alyssa Soucy, Elizabeth Pellecer Rivera, Natalie Siwek, Lucy Martin, Sarah Jackson, Gabrielle Venne, Sandra De Urioste-Stone.

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
