## [Decision Letter · Decision Letter 0]

21 Aug 2024

PONE-D-24-14900Opportunities and challenges within green spaces during COVID-19: Perspectives of visitors and managers in Maine, USAPLOS ONE

Dear Dr. Soucy,

Thank you for submitting your manuscript to PLOS ONE. After careful consideration, we feel that it has merit but does not fully meet PLOS ONE’s publication criteria as it currently stands. Therefore, we invite you to submit a revised version of the manuscript that addresses the points raised during the review process. The third reviewer has not yet submitted their review report. However, I believe it would be beneficial to forward the current reviews to you in order to facilitate the review process. Should a late review be received, it will be sent to you.

We look forward to receiving your revised manuscript.

Kind regards,

Mario Soliño

Academic Editor

PLOS ONE

Journal Requirements:

"This work was supported by the National Science Foundation under the Grants No. 1824961, 1828466, 1849802; the USDA National Institute of Food and Agriculture, McIntire Stennis project number #ME0-42017 through the Maine Agricultural & Forest Experiment Station; and by a University of Maine Research Reinvestment Fund Rural Health and Wellbeing Grand Challenge grant."

3. We note that your Data Availability Statement is currently as follows: All relevant data are within the manuscript and its Supporting Information files

Reviewers' comments:

Reviewer's Responses to Questions

**Comments to the Author**

1. Is the manuscript technically sound, and do the data support the conclusions?

Reviewer #1: Yes

Reviewer #2: Yes

2. Has the statistical analysis been performed appropriately and rigorously? 

Reviewer #1: Yes

Reviewer #2: N/A

3. Have the authors made all data underlying the findings in their manuscript fully available?

Reviewer #1: Yes

Reviewer #2: Yes

4. Is the manuscript presented in an intelligible fashion and written in standard English?

Reviewer #1: Yes

Reviewer #2: Yes

5. Review Comments to the Author

Reviewer #1: The paper is well written, adequately describes methods and results, had presents implications for managers.

My only concern is that not all the concepts used in the research are fully detailed in the literature review (e.g. place attachment is a complex topic, that needs more explanation, if the results related to identity and dependence are to be useful to readers.)

Reviewer #2: General Comments

The paper is well-structured and well-written, focusing on the effects of the COVID-19 pandemic on green areas in Maine, USA. The study employs a qualitative methodology, comparing different projects to understand how people, due to the pandemic, changed their ways of interacting with the green areas in the case study analyzed. It aims to explore the opportunities and challenges related to changing customs in the use of green areas, particularly the influence of COVID-19 on these changes in activities, perceptions, and uses within the case study. This topic is both interesting and necessary. However, the paper could benefit from a more in-depth examination of certain aspects to enhance the robustness of the results and reinforce the research gap and the importance of conducting similar studies. Below, I provide some comments and suggestions that can help to improve the paper, in case the authors want to consider them.

Specifically, I recommend that the authors expand the Introduction section to include more background on the research related to COVID-19 and the importance of green areas. Recent studies, in fact, have explored this topic from different perspectives. Additionally, it would be beneficial to clearly specify the research gap and the innovative contributions this article makes to the literature, differentiating it from other related studies. This will help to increase the robustness and novelty of the current study by clarifying its unique contributions compared to other research. Currently it is not so clear to me reading the text the novelty that the paper brings if we compare it with other literature on the same topic. The Introduction and Conclusions are likely good sections in which to highlight these aspects in more detail.

Specific comments

- Introduction – Literature Review section. In lines 92-95 and 100-102, the authors state that people have positive and negative experiences in green areas due to various factors, but do not go into detail. Can you provide examples or further elaborate on how people perceive green spaces differently according to the literature? What roles do values and socio-cultural or individual factors play in influencing these experiences?

- Line 132-135. Can you further explain in the text what causes an increase or decrease? What are the most common factors you found in the literature?

- Line 178. Figure 1. I think adding a figure of the green areas in the case study analyzed, with their distribution, different typologies, or some representative photos, could help the reader better understand the context being analyzed and the results obtained later.

- Line 186. Are you referring to visitors, local citizens, or tourists? Is the sense of place the same for people who don't live in the area compared to local people? Are you considering these distinctions, or are you talking about all types of visitors in these green spaces? It would be better to specify this in the text.

- Line 234. Can you add more information on how you triangulated the participants and methods? This could help increase the ease for readers who want to replicate the same methodology.

- Line 285-289. Can you provide further details about how you identified this 3 main category and what criteria helped you define them?

- Line 328-329. What type of “ecological damage” or “ecological worries” are you referring to?

- Line 402-404. Can you further elaborate on the concept of "attachment to the state"? What are you referring to? I think it can help the reader to better understand the concept of sense of place, identity, and attachment of you case study, as well as helping to extrapolate it more generally.

- Line 414. Table 3. It seems that the terms "Place Identity" and "Place Dependence" appear here with great importance, but their meanings have not been explained in detail in the text. I think lines 187-189 would be a good place to add a paragraph briefly explaining how the authors understand these two concepts within the context of their study, as these terms can have completely different meanings depending on the socio-cultural factors of the context being analyzed.

- Line 439-440. This reflection and suggestion is valid for all the results and discussions presented in the article. Is this perception different compared to the pre-COVID period? How was the general sense of attachment of people in your case study before the COVID-19 pandemic? Are there any differences? Could you further elaborate on this comparison between pre- and post-pandemic? How has the sense of place and identity of people changed? I think that giving more details could help to underline the innovative insights that have come to light with this study and give it greater robustness

- Line 591-593. You should discuss more about the values, norms, and perceptions of the people, as individual and community factors, in the case study analyzed, both as individual and community factors, in the case study analyzed, not just referring to other literature. Could you provide some concrete examples related to your results? This will strengthen the connection between results and discussion sections.

- Line 721. Could you further elaborate on the importance of local values and socio-cultural factors in influencing people's changing attitudes and perceptions regarding the use of green areas? How can these factors could influence the results?

- In the conclusions section, I would recommend that the authors to further emphasize the novelty of their study, distinguishing it from other literature that studies similar aspects related to COVID-19 and the change in people's perceptions and attitudes towards green areas.

6. PLOS authors have the option to publish the peer review history of their article (what does this mean? ). If published, this will include your full peer review and any attached files.

**Do you want your identity to be public for this peer review?** For information about this choice, including consent withdrawal, please see our Privacy Policy .

Reviewer #1: No

Reviewer #2: **Yes: ** Enrica Garau

---

## [Author Response · Author response to Decision Letter 1]

20 Feb 2025

PONE-D-24-14900

Opportunities and challenges within green spaces during COVID-19: Perspectives of visitors and managers in Maine, USA

PLOS ONE

Dear Dr. Soucy,

Thank you for submitting your manuscript to PLOS ONE. After careful consideration, we feel that it has merit but does not fully meet PLOS ONE’s publication criteria as it currently stands. Therefore, we invite you to submit a revised version of the manuscript that addresses the points raised during the review process.

The third reviewer has not yet submitted their review report. However, I believe it would be beneficial to forward the current reviews to you in order to facilitate the review process. Should a late review be received, it will be sent to you.

● A rebuttal letter that responds to each point raised by the academic editor and reviewer(s). You should upload this letter as a separate file labeled 'Response to Reviewers'.

● A marked-up copy of your manuscript that highlights changes made to the original version. You should upload this as a separate file labeled 'Revised Manuscript with Track Changes'.

● An unmarked version of your revised paper without tracked changes. You should upload this as a separate file labeled 'Manuscript'.

We look forward to receiving your revised manuscript.

Kind regards,

Mario Soliño

Academic Editor

PLOS ONE

Journal Requirements:

Response: Thank you for including the style guide, we have made the necessary changes to manuscript formatting to meet PLOS ONE’s style requirements.

"This work was supported by the National Science Foundation under the Grants No. 1824961, 1828466, 1849802; the USDA National Institute of Food and Agriculture, McIntire Stennis project number #ME0-42017 through the Maine Agricultural & Forest Experiment Station; and by a University of Maine Research Reinvestment Fund Rural Health and Wellbeing Grand Challenge grant."

Response: Thank you for highlighting this need, we have added the above statement to the manuscript.

3. We note that your Data Availability Statement is currently as follows: All relevant data are within the manuscript and its Supporting Information files

Response: We have included them as Supplemental Information individual files for each of the three quantitative studies containing relevant data sets.

Response: We have modified the ethics statement to include “The University of Maine Institutional Review Board (IRB) for research…”

Response: We have checked all the references for accuracy, as well as reformatted the reference style to meet journal guidelines.

Reviewers' comments:

Reviewer's Responses to Questions

Comments to the Author

1. Is the manuscript technically sound, and do the data support the conclusions?

Reviewer #1: Yes

Reviewer #2: Yes

2. Has the statistical analysis been performed appropriately and rigorously?

Reviewer #1: Yes

Reviewer #2: N/A

3. Have the authors made all data underlying the findings in their manuscript fully available?

Reviewer #1: Yes

Reviewer #2: Yes

4. Is the manuscript presented in an intelligible fashion and written in standard English?

Reviewer #1: Yes

Reviewer #2: Yes

5. Review Comments to the Author

Reviewer #1: The paper is well written, adequately describes methods and results, had presents implications for managers.

My only concern is that not all the concepts used in the research are fully detailed in the literature review (e.g. place attachment is a complex topic, that needs more explanation, if the results related to identity and dependence are to be useful to readers.)

Response: Thank you for bringing up this important point regarding further elaboration of sense of place/place attachment. First, we moved the definitions of sense of place/place attachment from study site description to the section on “People and green spaces.” We further elaborated within this paragraph to offer more context in regards to how sense of place is linked with green spaces, including the following: “Sense of place significantly influences not only individuals’ experiences within green spaces, but also their trust in the management of these areas [57], motivations for recreation visits and satisfaction with their visits [58], and willingness to travel to or advocate for certain locations [59]. For example, studies have shown that strong place identity fosters trust in park authorities and stewardship, while high place dependence enhances satisfaction and loyalty to a specific green space [60,61].”

Reviewer #2: General Comments

The paper is well-structured and well-written, focusing on the effects of the COVID-19 pandemic on green areas in Maine, USA. The study employs a qualitative methodology, comparing different projects to understand how people, due to the pandemic, changed their ways of interacting with the green areas in the case study analyzed. It aims to explore the opportunities and challenges related to changing customs in the use of green areas, particularly the influence of COVID-19 on these changes in activities, perceptions, and uses within the case study. This topic is both interesting and necessary. However, the paper could benefit from a more in-depth examination of certain aspects to enhance the robustness of the results and reinforce the research gap and the importance of conducting similar studies.

Below, I provide some comments and suggestions that can help to improve the paper, in case the authors want to consider them.

Specifically, I recommend that the authors expand the Introduction section to include more background on the research related to COVID-19 and the importance of green areas. Recent studies, in fact, have explored this topic from different perspectives. Additionally, it would be beneficial to clearly specify the research gap and the innovative contributions this article makes to the literature, differentiating it from other related studies. This will help to increase the robustness and novelty of the current study by clarifying its unique contributions compared to other research. Currently it is not so clear to me reading the text the novelty that the paper brings if we compare it with other literature on the same topic. The Introduction and Conclusions are likely good sections in which to highlight these aspects in more detail.

Response: We appreciate your careful review of our manuscript, and identification of a need to further articulate the novelty of our work. Underscoring the innovativeness of our work was a key focus of the revision. First, we expanded the literature review to include more background on COVID-19 influences on recreation to provide further context on closures, restrictions, and the importance of green spaces for social interactions (see “Emergence of COVID-19 in the context of green spaces”). Additionally, we added the following paragraph at the end of the introduction to further situate our work:

“This case study provides a novel and innovative contribution to research on COVID-19 and green spaces by examining diverse perceptions and pandemic-related impacts through the lens of place attachment within a rural region. Existing studies have highlighted changes in visitation during COVID-19 and its benefits for mental and physical health; however, this research distinguishes itself by focusing on Maine, a state characterized by its expansive green spaces and its reliance on tourism. Maine offers a compelling lens for understanding how place attachment—a sense of identity, connection, and emotional bond to specific locations—influenced responses to the pandemic and patterns of use and stewardship. By incorporating diverse perspectives from green space managers and visitors, this study captures the layered meanings and values associated with these spaces during the pandemic, offering transferable insights for other rural, tourism-dependent regions. Furthermore, the study’s snapshot approach documents specific moments in time, complementing longitudinal research by revealing how place attachment evolved under crisis conditions [20]. These findings underscore the importance of integrating place-based values into adaptive management strategies to sustain green spaces into the future. This research provides a critical framework for policymakers and resource managers to balance ecological preservation with public access, while fostering meaningful connections that enhance both individual well-being and long-term conservation goals.”

Specific comments

- Introduction – Literature Review section. In lines 92-95 and 100-102, the authors state that people have positive and negative experiences in green areas due to various factors, but do not go into detail. Can you provide examples or further elaborate on how people perceive green spaces differently according to the literature? What roles do values and socio-cultural or individual factors play in influencing these experiences?

Response: This is an important point and therefore we added additional context to clarify on the positive and negative experiences associated with green spaces. For example, we added the following points to the first paragraph under “People and green spaces”: However, experiences, historical relationships with land (i.e., exclusion from land), sociocultural practices, family roles, sense of belonging, experiences with discrimination, and resource availability informs how groups engage with the outdoors and the types of experiences they derive [32,33]. Individuals may feel anxious or uncertain concerning the use of green spaces, which can ultimately inhibit the benefits of green spaces and make people feel unwelcomed [34,35]. In other words, the benefits of outdoor recreation have not been equitably shared across all communities [33]. Factors such as race and ethnicity, economic status, and gender have significantly impacted the extent to which structural, interpersonal, and intrapersonal barriers influence

---

## [Editor Report · Decision Letter 1]

25 Feb 2025

Opportunities and challenges within green spaces during COVID-19: Perspectives of visitors and managers in Maine, USA

PONE-D-24-14900R1

Dear Dr. Soucy,

We’re pleased to inform you that your manuscript has been judged scientifically suitable for publication and will be formally accepted for publication once it meets all outstanding technical requirements.

Kind regards,

Mario Soliño

Academic Editor

PLOS ONE

Additional Editor Comments (optional):

Authors have satisfactorily accomplished with all the previous reviewers' comments.
---

## [Editor Report · Acceptance letter]

PONE-D-24-14900R1

PLOS ONE

Dear Dr. Soucy,

I'm pleased to inform you that your manuscript has been deemed suitable for publication in PLOS ONE. Congratulations! Your manuscript is now being handed over to our production team.

Kind regards,

on behalf of

Dr. Mario Soliño

Academic Editor

PLOS ONE